# Soybean Plants Exposed to Low Concentrations of Potassium Iodide Have Better Tolerance to Water Deficit through the Antioxidant Enzymatic System and Photosynthesis Modulation

**DOI:** 10.3390/plants12132555

**Published:** 2023-07-05

**Authors:** Jucelino de Sousa Lima, Otávio Vitor Souza Andrade, Leônidas Canuto dos Santos, Everton Geraldo de Morais, Gabryel Silva Martins, Yhan S. Mutz, Vitor L. Nascimento, Paulo Eduardo Ribeiro Marchiori, Guilherme Lopes, Luiz Roberto Guimarães Guilherme

**Affiliations:** 1Department of Biology, Institute of Natural Sciences, Federal University of Lavras (UFLA), Lavras 37200-900, MG, Brazil; sousajucelino12@gmail.com (J.d.S.L.); otaviovsandrade@gmail.com (O.V.S.A.); vitor.nascimento@ufla.br (V.L.N.); paulo.marchiori@ufla.br (P.E.R.M.); 2Department of Soil Science, School of Agricultural Sciences, Federal University of Lavras (UFLA), Lavras 37200-900, MG, Brazil; leonidas.santos2@estudante.ufla.br (L.C.d.S.); evertonmoraislp@gmail.com (E.G.d.M.); gabryel.martins1@estudante.ufla.br (G.S.M.); yhan.mutz@ufla.br (Y.S.M.); guilherme.lopes@ufla.br (G.L.); 3Department of Food Science, School of Agricultural Science, Federal University of Lavras (UFLA), Lavras 37200-900, MG, Brazil

**Keywords:** iodine, abiotic stress tolerance, antioxidant defense

## Abstract

Water deficit inhibits plant growth by affecting several physiological processes, which leads to the overproduction of reactive oxygen species (ROS) that may cause oxidative stress. In this regard, iodine (I) is already known to possibly enhance the antioxidant defense system of plants and promote photosynthetic improvements under adverse conditions. However, its direct effect on water deficit responses has not yet been demonstrated. To verify the efficiency of I concerning plant tolerance to water deficit, we exposed soybean plants to different concentrations of potassium iodide (KI) fed to pots with a nutrient solution and subsequently submitted them to water deficit. A decline in biomass accumulation was observed in plants under water deficit, while exposure to KI (10 and 20 μmol L^−1^) increased plant biomass by an average of 40%. Furthermore, exposure to KI concentrations of up to 20 μM improved gas exchange (~71%) and reduced lipid peroxidation. This is related to the higher enzymatic antioxidant activities found at 10 and 20 μM KI concentrations. However, when soybean plants were properly irrigated, KI concentrations greater than 10 μM promoted negative changes in photosynthetic efficiency, as well as in biomass accumulation and partition. In sum, exposure of soybean plants to 10 μM KI improved tolerance to water deficit, and up to this concentration, there is no evidence of phytotoxicity in plants grown under adequate irrigation.

## 1. Introduction

The rise in the world’s population—which is expected to reach ~9.5 billion by 2050—requires continuous increases in crop production to ensure food security [1]. However, climate change is becoming one of the biggest threats to agricultural sustainability worldwide and, consequently, to the food supply chain, mainly due to increased temperatures that generate greater evapotranspiration and water scarcity [2]. Indeed, by 2050, water scarcity worldwide will create severe problems for plant growth and, consequently, yield, thus threatening global food security [3].

Under water deficit, one of the most typical consequences in leaves is the increase in reactive oxygen species (ROS) inducing lipid peroxidation [4,5]. Excessive accumulation of ROS causes the oxidation of nucleic acids, proteins, and lipids, which ultimately causes cellular dysfunction [6,7]. The excess production of ROS also causes disruption to the electron transport chain in the chloroplasts, as well as suppresses carbon fixation by inactivating the enzymes in the Calvin–Benson–Bassham cycle, resulting in a reduction in the photosynthetic rate and, consequently, in yield [6,8,9,10].

Among relevant crops cultivated globally, soybean (*Glycine max* (L.) Merrill) is one of the most economically important in the world, being the main cultivated oilseed and also an important protein source. Soybean is used by agroindustries in the production of vegetable oil as well as human and animal feed, in addition to being an alternative source for the manufacture of biofuels [11]. Brazil is the largest soybean producer in the world; data from the 2022/23 harvest indicate a planted area of 43,834.4 thousand ha^−1^, a production of 154,810.7 thousand tons, and an average yield of 3.542 kg ha^−1^, reaching historical records for planting area, production, and yield [12].

These high yields depend on several factors, such as water availability, which can be a crucial limitation to the expression of a crop’s yield potential, especially in years with uneven distribution and a low volume of precipitation in non-irrigated crops [13]. Water deficit stress causes a decrease in the yield and quality of soybean, so strategies to mitigate this negative effect are needed [14]. The flowering stage is one of the most critical periods in terms of water supply, and a deficit at this time, depending on the intensity, can lead to premature flower drop, with a consequent decrease in the number of pods, one of the main productive components of this culture [13].

The exogenous application of beneficial elements has become an important strategy to partially mitigate the adverse effects of water deficit in plants. These elements are not essential for survival, but they increase plant biomass and yield by stimulating various growth-promoting pathways, as well as helping to alleviate abiotic and biotic stresses [15,16,17]. Plant cells are generally protected by a complex antioxidant system, which may be enzymatic, including superoxide dismutase (SOD), ascorbate peroxidase (APX), catalase (CAT), and other peroxidases. SOD is the first line of defense against ROS, dismutating the radical superoxide (O_2_^−^) into peroxide (H_2_O_2_) and O_2_, while the others prevent the formation of hydroxyl radicals (-OH), the most toxic and reactive radicals, which can react indiscriminately with all macromolecules [18]. Other works highlight the ubiquitous role of antioxidant enzymes in mitigating different abiotic stresses [19,20]. Recently, it was demonstrated that the exogenous application of iodine (I) reinforces the antioxidant capacity of lettuce, soybean, and tomato plants by stimulating the activity of the main ROS detoxifying enzymes, i.e., superoxide dismutase (SOD), ascorbate peroxidase (APX), catalase (CAT), and guaiacol peroxide (POD) [18,21,22,23]. Higher antioxidant enzymatic activity in plants reduces oxidative damage promoted by ROS accumulation under water deficit [24,25]. Another point to highlight is the role of I in being covalently linked to at least 82 different proteins in the leaves and roots of *Arabidopsis thaliana*, and its presence in micromolar concentrations in a nutrient solution, which increased the accumulation of tomato plant biomass [26].

Iodine is currently classified as a non-essential element, but it is beneficial to plants, with the ability to increase tolerance to some types of stress, such as saline [16,26,27]. However, little is known about this beneficial effect on water deficit and on which mechanisms I can act to mitigate the resultant stress. Simultaneous evidence suggests that I can be used as a plant stimulant similar to silicon (Si), selenium (Se), and sodium (Na) [16]. Although there are indications of the beneficial effect of iodine on plants under conditions of abiotic stress (salinity and heavy metals, among others), there are no studies dealing with the effect of this element on plants under conditions of water deficit. Therefore, this study hypothesizes that exposure of soybean plants to I can improve plant tolerance to water deficit with stimulation of the enzymatic antioxidant system. Thus, the objective of the present study was to determine whether the application of I can improve the response to water deficit in soybean plants, increasing the enzymatic activity of the antioxidant system as well as the photosynthetic efficiency, thus improving the productive capacity of plants under water stress.

## 2. Results

### 2.1. Plant Growth

The water deficit significantly affected (*p* < 0.05) the growth variables of soybean plants. However, the application of potassium iodide (KI) improved these variables under water deficit in a dose-dependent manner (Figure 1). Under good irrigation conditions, an average reduction of 56% in total dry mass (TDM) was observed when plants were supplemented with 40 µM of KI compared with the other treatments (Figure 1A). In addition, there was a 39% reduction in shoot dry mass (SDM) at the concentrations of 20 and 40 µM, compared with the other treatments (Figure 1A,B). However, under water deficit, mean increments of 43% in TDM and 37% in SDM were verified for plants treated with 10 and 20 µM I.

When plants were treated with 20 µM of KI under conditions of full irrigation, the root dry mass (RDM) and root:shoot ratio (RDM:SDM) increased by 16% and 30%, respectively, compared with plants without KI (Figure 1C,D). It was also possible to observe that, under optimal irrigation, the root:shoot ratio improved by 23% in the treatment with 40 µM of KI compared with plants without KI, while under water suppression, no significant influence was observed between treatments with KI for RDM and RDM:SDM. For the water deficit tolerance index (WDTI), it was verified that KI concentrations did not significantly influence the plants with adequate irrigation (Figure 1E). However, under conditions of water suppression, the concentrations of 10 and 20 µM I increased the index by an average of 31% compared with the other treatments.

### 2.2. Leaf Gas Exchange

Leaf gas exchange variables were significantly influenced by water deficit and KI application (*p* < 0.05). Water deficit reduced the CO_2_ assimilation rate (*A*), stomatal conductance (*g_s_*), transpiration (*E*), and carboxylation efficiency (*CE*) irrespective of the applied KI concentration. On the other hand, under conditions of water deficit, an average increase of 86% and 71% in *A* was observed when plants were treated with 10 and 20 µM I, respectively, whereas, in well-watered plants, a 70% reduction in *A* was observed with the application of 40 µM of KI (Figure 2A). When comparing all treatments under water deficit, the application of 10 µM I promoted higher values of *A*, which reached 9.33 µmols CO_2_ m^−2^ s^−1^.

The *g_s_* had an average increase of 47% in the concentration of 10 µM KI compared with watered plants that received 10 and 40 µM KI (Figure 1B). Under water deficit, no significant difference was found for *g_s_* between the treatments. For *E* under conditions of adequate irrigation, the highest value observed was in the treatment with 20 µM of KI (6.48 mmol H_2_O m^−2^ s^−1^) followed by the treatment with 10 µM (5.40 mmol H_2_O m^−2^ s^−1^) and without KI application (2.99 mmol H_2_O m^−2^ s^−1^), where all were significantly different from the KI 40 µmol treatment (1.69 mmol H_2_O m^−2^ s^−1^) (Figure 2C). However, under water deficit, the concentration of 10 µM KI provided greater *E*, with an average increase of 73% when compared with the treatments of 0 and 40 µM KI. For the internal concentration of CO_2_ (*C_i_*), there was an increase of 16% for plants submitted to water deficit that did not receive the KI treatment when compared with the irrigated ones (Figure 1D). Under water deficit conditions, *C_i_* was reduced by 14% with the application of 20 µM I compared with plants without the use of KI.

Regarding water use efficiency (*WUE*), there was an average reduction of 42% when soybean plants were supplemented with KI under adequate irrigation. On the other hand, under conditions of water deficit, concentrations of 10 and 20 µM KI increased *WUE* by 49% when compared with the other treatments, while for *CE*, an average reduction of 77% was observed when applying 40 µM KI under adequate irrigation conditions. When subjected to water deficit, increases of 86% and 73% in *CE* were observed for plants supplemented with 10 and 20 µM I, respectively, in comparison with the other treatments. It can also be highlighted that under water deficit, the plants treated with 10 µM KI had the highest *CE* (0.26 µmol CO_2_ m^−2^ s^−1^ Pa^−1^) among all the evaluated treatments.

### 2.3. Oxidative Damage

Malondialdehyde (MDA) content was significantly increased (*p* < 0.05) with water deficit irrespective of the applied KI dose (Figure 3A). However, under water deficit conditions, an average reduction of 18% in MDA was observed when plants were exposed to 10 and 20 µM I compared with plants grown without KI application.

Similarly, water deficit and KI concentrations significantly (*p* < 0.05) affected H_2_O_2_ content (Figure 3B). For adequate irrigation, treatment with 40 µM KI reduced the concentration of hydrogen peroxide (H_2_O_2_) by 28%, compared with plants without KI. However, under water deficit, a 46% increase was observed when the plants were exposed to 20 µM I compared with plants without KI under the same irrigation conditions. It is worth highlighting that when treated with KI (10, 20, and 40 µM) and subjected to water deficit, H_2_O_2_ increased by ~49% compared with irrigated plants cultivated with the same KI concentrations.

### 2.4. Antioxidant Enzymatic Activity

The antioxidant enzymatic activity was significantly influenced (*p* < 0.05) by both water deficit and KI concentrations (Figure 4). For adequate irrigation, there was an increase in SOD activity—corresponding to 53% and 70%—when plants received 10 and 20 µM KI, respectively, compared with the other treatments (Figure 4A). It can be highlighted that the highest SOD activity was found in the treatment with 20 µM (9.70 U mg^−1^ protein min^−1^). Under water deficit, the activity of this enzyme increased by 69% when plants did not receive KI compared with their irrigated counterparts. However, when plants suffering water deficit were treated with 20 µM KI, an increase of 36% was observed compared with the plants without KI and with 10 µM, and 87% when compared with the treatment with 40 µM KI.

Regarding APX activity, plants supplemented with 40 µM KI and under optimal irrigation exhibited an average increase of 76% in relation to plants without and with 10 µM KI (Figure 4B). However, under water deficit, an average increase of 68% was observed for concentrations of 10 and 20 µM KI compared with the other treatments.

Under well-irrigated conditions, exposure to I increased the activity of CAT by an average of 90% compared with plants without KI. Under conditions of water deficit, there was an increase of 94% when plants received the treatment with 10 and 40 µM KI and 97% in the treatment with 20 µM when compared with the plants without KI. Notably, the highest CAT activity under water deficit conditions was observed in plants supplemented with 20 µM KI (877.29 µmols H_2_O_2_ mg^−1^ protein^−1^) (Figure 4C).

POD activity increased between 35% and 65% when plants were treated with 20 and 40 µM I and adequately irrigated, relative to plants without KI (Figure 4D). In the same irrigation condition, the treatment with 40 µM KI provided greater enzyme activity (5.51 µmols tetraguaiacol mg^−1^ protein min^−1^). However, under water stress, the highest POD activity was related to the application of 20 µM KI (5.35 µmols tetraguaiacol mg^−1^ protein min^−1^), which improved the activity of this enzyme by 42% compared with plants without KI and with 40 µM, in addition to 26% compared with treatment with 10 µM KI.

### 2.5. Proline

Soybean plants had significant changes (*p* < 0.05) in proline content in relation to the two studied factors (water deficit and KI application) (Figure 5). Under adequate irrigation, average increases of 55% were observed when plants were grown with 20 and 40 µM I compared with plants without and with 10 µM KI. Proline contents increased under deficient irrigation compared with their adequate irrigation condition counterparts, regardless of the applied KI concentration. It is noteworthy that the highest concentration of proline in plants under water deficit was found in plants without KI (2.56 mg g^−1^ FW).

### 2.6. Multivariate Analysis

A principal component analysis (PCA) was performed using the morphological, physiological, and biochemical characteristics of soybean cultivated with adequate irrigation. The first two principal components (PC1 and PC2) presented 66.53% of the total data variance (Figure 6). The biplot for optimal irrigation conditions revealed a strong relationship between the treatment without KI and SDM and *WUE*. These two variables correlated positively and negatively with APX and POD activities, which were positively correlated with the highest KI dose (40 µMKI).

On the other hand, the treatment with 10 µM I tended to favor TDM, SDM, *A*, *CE*, *g_s_*, and H_2_O_2_, which had a positive correlation between themselves and a negative correlation with POD. Furthermore, TDM, SDM, *A*, and *CE* were negatively correlated with APX. Lastly, *A*, *CE*, and *g_s_* also had a negative correlation with MDA. MDA tended to be favored by the 40 µM KI concentration. Further, MDA correlated negatively with SOD and *E*, which were positively correlated and tended to be favored by the treatment with 20 µM KI. In addition, RDM also tended to be favored by the 20 µM KI concentration and was positively correlated with SOD and *E*.

The cumulative variance in PC1 and PC2 under water deficit conditions was 68.43% (Figure 7). The concentration of 10 µM KI tended to be positively correlated with most of the analyzed variables, including WDTI, SMD, TDM, *A*, *E*, *CE*, and APX, which had positive correlations with each other. Furthermore, such concentration also indicated a favoring in RDM correlated positively with WDTI, TDM, and SDM. The treatment without KI, on the other hand, tended to correlate with proline, *C_i_*, and *g_s_*, which correlated positively with each other and negatively with *WUE*. *WUE* presented a correlation with the treatment with 20 µM KI, which also favored POD and H_2_O_2_, with a positive correlation between all the mentioned variables. Furthermore, all these variables mentioned above for the 20 µM KI treatment had a negative correlation with MDA, which is favored by treatments without KI and with 40 µM KI. It is also noteworthy that MDA had a negative correlation with most variables that tended to be favored by the 10 µM KI concentration (WDTI, TDM, SDM, RDM, *A*, *E*, *CE*, and APX).

## 3. Discussion

Stress in plants due to water deficit reduces their growth due to cell turgor reduction, followed by damages in the photosynthetic process, causing increases in oxidative damage [4,5,28]. However, these alterations promoted by stress may not necessarily harm plant development since supplementation of exogenous compounds or elements with beneficial properties can increase plant tolerance levels to specific stressful conditions [24,29,30]. In the case of water deficit stress, plants can circumvent damage by improving their antioxidant capacity and adjusting gas exchange [24,25].

In the present study, a reduction in growth due to water deficit was observed in the accumulation of biomass in soybean plants (Figure 1). A decrease in soybean biomass resulted mainly from a reduction in the efficiency of photosynthesis and an increase in lipid peroxidation due to oxidative stress, indirectly indicated by the accumulation of MDA [25]. Exposure of soybean plants to a low concentration of KI (10 and 20 µM) increased the antioxidant enzymatic activity, consequently reducing the MDA content, which improved photosynthetic efficiency, thus promoting greater tolerance to water suppression (WDTI). The effect of I related to water deficit is still little reported in the literature; however, reports point to its beneficial effect on the germination and growth of *Carthamus tinctorius* L. subjected to water deficit [31]. In addition, the effect of this element was also reported for other abiotic stresses, such as salinity [22,26,32] and heavy metals [21].

Our results indicated that exposure to iodine showed a capacity for photosynthetic improvement in plants under water deficit when concentrations of 10 and 20 μM I were applied, which is reflected by the increase in biomass in these respective treatments. A study reported that iodinated proteins could improve photosynthesis and defense responses in plants [26]. The authors pointed out that most of the iodinated proteins in the shoot were connected with chloroplasts and were involved in photosynthetic processes. Corroborating our study findings, the photosynthetic rate of lettuce was shown to increase by ~20% due to the application of I, in the form of IO_3_^−1^, in concentrations of 20, 40, and 80 μM [18]. Furthermore, as already mentioned, the greater antioxidant protection at concentrations of 10 and 20 μM may also ensure greater photosynthetic performance in plants grown under water stress by effectively reducing the damage caused by ROS, thus balancing ROS synthesis and signaling to provide greater stress tolerance [33]. The excessive accumulation of ROS promoted with water deficit can cause oxidative damage to the electron transport chain, increase lipid peroxidation in chloroplasts and mitochondria, inactivate enzymatic activity, directly oxidize proteins and nucleic acids, and, finally, decrease photosynthesis and crop production [24,34].

Iodine is considered a potential antioxidant in plants since the antioxidant capacity of plants is positively correlated with the amount of I [35,36,37,38]. Furthermore, studies on soybeans and lettuce indicated that the application of I at concentrations of 20, 40, and 80 μM increases the enzymatic activities of SOD, APX, and CAT [18,21]. Evaluating different compounds containing I, a study reported decreases in POX (guaiacol peroxidase) activity for tomato plants exposed to increasing concentrations of I, with the exception of the highest doses (i.e., 50 mM I) [39]. The results cited above corroborate our findings. Furthermore, under water deficit, our results indicated greater APX, POD, and CAT activities, mainly when the plants were exposed to 10 and 20 μM KI. Exposure of soybean plants to 10 and 20 μM KI provided greater antioxidant enzymatic activity, which consequently may have guaranteed less oxidative stress, as observed by the lower MDA content. A high MDA content indicates a high degree of lipid peroxidation in the plant membrane and a state of oxidative stress [40]. This occurs when ROS levels are exceeded, causing lipid peroxidation, protein oxidation, and enzyme inactivation, which can even lead to cell death [41]. A high-performance antioxidant system contributes to lowering oxidative stress and helps plant growth. In *A. thaliana*, iodinated proteins were identified and confirmed as belonging to POD class III [26]. This may relate to ROS oxidizing I^−^, which leads to the preferential iodization of close proteins, including enzymes from the antioxidant system [42].

An interesting finding from our study was the higher H_2_O_2_ content in plants grown under water deficit and exposed to 10 and 20 μM I, which were the treatments that provided the highest growth in this irrigation condition. H_2_O_2_ is a molecule capable of reacting with cellular components interfering with normal physiological and metabolic functions and unbalancing the cellular redox homeostasis [43,44]. However, H_2_O_2_ also plays important roles in plant development and physiological processes, including flowering [45], root system development [46,47,48], regulation of stomatal opening [49], and many others. Furthermore, the levels found in our study were below those generally considered harmful for some cellular compartments (basal H_2_O_2_ level of 5–15 μmols) [50,51]. The low levels found in plants under water deficit and without KI may have been related to the H_2_O_2_ breakdown into other molecules as its dismutation into hydroxyl radicals (^−^OH). These radicals, in turn, are more reactive than H_2_O_2_, leading to lipid peroxidation, and thus may explain the greater concentration of MDA in their respective treatment [52]. Thus, it was observed that in the plants treated with 10 and 20 μM KI, which had lower MDA content and higher H_2_O_2_ values, the dismutation products of H_2_O_2_ may have been H_2_O and O^2−^ due to the greater antioxidant enzymatic activity in these treatments, enabling higher growth in conditions of water deficit [53,54]. This hypothesis represents a potential mechanism that suits this study’s findings but needs more results to be confirmed.

Despite the beneficial effect of KI observed under the water stress condition, the treatment with the highest KI dose (40 μM) proved to be harmful to the plants since they had a higher MDA content and lower growth. These results indicate a possible toxicity of I at this concentration in plants subjected to water deficit, a fact related to the pro-oxidative effect of I at high concentrations [37,54]. The phytotoxic effect of I was also seen in the present study under conditions of adequate irrigation, as there was a change in the root:shoot ratio with the concentration of 20 μM I, and a reduction in the accumulation of biomass in plants treated with 40 μM I. In agreement with our results, Blasco et al. [18] observed that the photosynthetic rate of lettuce significantly decreased when the I concentration exceeded 40 μM, causing lower growth and biomass accumulation. In a study on basil, Incrocci et al. [37] observed a decline in plant height, leaf area, and biomass accumulation when KI concentrations were greater than 50 μM. Kato et al. [55] reported a 26% decrease in the shoot length of rice treated with 25 μM I. Thus, it is observed that the beneficial effect of I depends on the concentration used, as in excess, I can potentially induce oxidative stress [38].

Although I accumulates to some extent in plants, it can reduce oxygen-containing free radicals and oxygen-containing molecules such as superoxide anions, H_2_O_2_, and ^−^OH, [56]. Additionally, the formation of I proteins or the involvement of I as an inducing factor in protein synthesis may be the cause of I toxicity [54]. Thus, although plants under water deficit treated with 20 μM KI had higher activities of enzymes such as SOD, CAT, and POD and good accumulation of biomass, at this exposure concentration, I began to promote small disturbances in plants under adequate irrigation. Therefore, the concentration of 10 μM I would be the most appropriate exposure treatment, as it promotes a beneficial effect under conditions of stress due to water deficit, without having a harmful effect under conditions of adequate irrigation. It is crucial to highlight that the morphophysiological and biochemical variables analyzed also pointed to 10 μM I as the best treatment in a conjoint manner, as depicted with the PCA evaluation.

Lastly, although plants under water deficit have increased proline content, this osmolyte was not responsible for soybean plants’ increased tolerance induced using exposure to I. Thus, our findings suggest that the main process responsible for the increase in tolerance to water stress with the application of I was related to the increase in the production of antioxidant enzymes, which promoted better protection of the photosynthetic apparatus, thus allowing the accumulation of biomass in soybean plants under water deficiency.

## 4. Materials and Methods

### 4.1. Cultivation System, Experimental Design, and Treatments

Soybean plants were grown in pots with 1000 g washed sand, with one plant per pot. The dimensions of the pots were 15 × 9 × 9. The cultivation was carried out in a greenhouse located in the Experimental Area of the Plant Physiology Sector in the Biology Department at the Federal University of Lavras (UFLA) (21°14′45″ S, 44°59′59″ W; 920 m above sea level), southeastern Brazil. Plants were exposed to a mean temperature of 25 °C and an 11/13.5 h (winter/summer) photoperiod and were fertigated twice a week with 50 mL of Hoagland and Arnon’s nutrient solution [57]. The treatments added to the pots were arranged in a completely randomized design, with five replications of each treatment. The experiment was carried out in a 4 × 2 factorial scheme, corresponding to four concentrations of I added with a nutrient solution (0; 10; 20; 40 μmols L^−1^) and two irrigation conditions (with and without water deficit). The experiment had a total of 40 experimental units. In total, there were 13 applications of 50 mL of the solution containing KI during the experiment, making a total of 650 mL. In this way, we can say that there was an application of approximately 0.825, 1.65, and 3.3 g of I kg^−1^ of substrate in treatments of 10, 20, and 40 μM KI, respectively.

The treatments related to the application of I were carried out by adding KI to the nutrient solution starting 14 days after seed germination at stage V2 (two leaflet). Irrigation was suspended 60 days after germination (DAG) to subject the plants to water deficit, and when this was reached, immediate rehydration was performed. The water deficit was established by monitoring gas exchanges, i.e., when it reached a negative A value and an E value of less than 1 mmol H_2_O m^−2^ s^−1^ (Figure 8). These values were achieved on the fourth day after the suspension of irrigation. The mean values of A and E for each treatment are shown in Table 1.

The sample collection for the biochemical evaluations was carried out on the day the plants reached water deficit, and the analysis of gas exchanges was completed one day after rehydration. At 64 DAG, the experiment was completed, and the plants were collected for biomass evaluation, separating the plant into shoots and roots.

### 4.2. Biomass and WDTI

At the end of the experiment, the SDM, RDM, and TDM RDM:SDM were determined. To obtain dry mass, the tissues were dried at 70 °C in a forced-circulation oven until constant mass. In addition, the WDTI was calculated according to [58,59] using the following equation:$$WDTI=\left(\frac{total\;dry\;mass\;of\;control}{total\;dry\;mass\;of\;other\;treatments}\right)\times 10$$

### 4.3. Leaf Gas Exchange

As previously mentioned, analyses were performed one day after the rehydration of the plants using an infrared gas exchange analyzer (IRGA, model LICOR 6400, Li-COR Biosciences, Lincoln, NE, USA). Data collection was performed between 8 am and 10 am, and the following variables were evaluated: CO_2_ assimilation (*A*—μmol CO_2_ m^−2^ s^−1^), stomatal conductance (*g_s_*—mol H_2_O m^−2^ s^−1^), transpiration (*E*—mmol H_2_O m^−2^ s^−1^), and internal CO_2_ concentration (*C_i_*—μmol CO_2_ mol air^−1^). Based on the *A* a *E* results, water use efficiency estimates (*WUE* (μmol CO_2_ mmol^−1^ H_2_O)) was calculated as *A*/*E*. The carboxylation efficiency (*EC*—μmol CO_2_ m^−2^ s^−1^ Pa^−1^) was estimated for the through the results of *A* and *Ci* (*A*/*Ci*). Atmospheric CO_2_ inside the leaf chamber was maintained at 400 μmol CO_2_ mol air^−1^, irradiance at 1500 μmols m^−2^ s^−1^, and leaf temperature at 25 °C. The pre-established minimum time for stabilization of the readings was 120 s.

### 4.4. H_2_O_2_ and MDA Content

To evaluate the H_2_O_2_ and MDA content, 0.2 g of fresh material was collected and macerated in a mortar with liquid nitrogen, homogenized in 1500 μL of trichloroacetic acid (TCA), and centrifuged at 12,000× *g* for 15 min at 4 °C. The H_2_O_2_ content was determined by collecting the supernatant and then reading its absorbance at 390 nm in a medium composed of 10 mM potassium phosphate (pH 7.0), 45 μL of plant material extract, and 1 M potassium iodide [60].

The MDA quantification was performed according to Buege and Aust [61]. For this, 125 μL of the extraction supernatant was collected and pipetted into a 1500 μL microtube containing 250 μL of the following reaction medium: 0.5% thiobarbituric acid (TBA) and 10% TCA. The microtubes then were placed in a water bath at 95 °C for 30 min, and after that, the reaction was stopped using cooling on ice. Subsequently, 350 μL of the reaction medium was collected and pipetted into microplates and read using a spectrometer at 535 and 600 nm. The content of MDA was obtained according to the following equation:MDA = (A535 − A600)/(ξ.b)
where ξ: (molar extinction coefficient = 1.56 × 10^−5^ cm^−1^) and b: (optical length = 1). Lipid peroxidation was expressed in nmol (MDA) g^−1^ of fresh matter.

### 4.5. Antioxidant Enzymatic Activity

For the evaluation of SOD, APX, CAT, and POD enzymes, 0.2 g of fresh material was collected and then ground in liquid nitrogen with the subsequent addition of 1.5 mL of a buffer solution (0.1 mol L^−1^ of potassium phosphate pH (7.8), 0.1 mol L^−1^ EDTA (pH 7.0), 0.5 mol L^−1^, DTT, 0.1 mol L^−1^ PMSF, 1.0 mmol L^−1^ ascorbic acid, and 22.0 mg PVPP). Soon after, the suspension was centrifuged at 13,000× *g* for 10 min at 4 °C, and the supernatant was collected for analysis in a spectrophotometer (Epoch-BioTek, Miami, FL, USA).

The SOD activities in leaves and roots were determined by quantifying the inhibition in photoreduction in nitrobluetetrazolium (NBT), following the protocol devised by Beauchamp and Fridovich [62]. The reaction solution was prepared by mixing: (i) 75 µL of NBT; (ii) 20 µL of riboflavin; (iii) 130 mL of L-methionine; and (iv) 100 µL of Na_2_EDTA into a sodium phosphate buffer. Next, this solution (2.725 mL) was mixed with H_2_O (0.25 mL) and 50 µL enzyme extract (supernatant) into a glass beaker and was kept in the dark. A similar set of beakers was prepared and placed in light conditions of 4.000 lux for 15 min. The absorbances of the samples kept in the dark and illuminated were recorded at 560 nm using a spectrophotometer.

The activity of APX was determined using Nakano and Asada’s [63] methodology. Briefly, the reaction solution contained 100 µL ascorbate solution (10 mM), 100 µL H_2_O_2_ (30%), and 100 µL enzyme extract (supernatant) in 2.7 mL of sodium phosphate buffer. After a gentle shake, the absorbance was read at 290 nm with on time scan (0–60s) using a spectrophotometer.

The reaction solution for POD contained 100 µL 30 mM H_2_O_2_, 100 µL guaiacol, and 100 µL enzyme extract (supernatant) in 2.7 mL sodium phosphate buffer. While for the estimation of CAT activity, the same reaction solution used for POD (except guaiacol) was used. The absorbance of POD and CAT samples was observed on time scan (0–60 s) at 470 and 240 nm, respectively, using a spectrophotometer [64].

To calculate the specific activity of antioxidant enzymes, the total soluble protein content was determined using the enzymatic extraction method. The microplates first received 294 μL of Bradford’s solution [65] at a 1:5 dilution of the reagent. The readings were performed using an absorbance microplate reader (Epoch-BioTek) at a wavelength of 595 nm, and the results were obtained from a calibration curve with BSA.

### 4.6. Proline

Extraction and quantification were performed according to the methodology proposed by Bates et al. [66]. Briefly, 0.2 g of plant material was macerated in 3% sulfosalicylic acid (10 mL) followed by stirring for 60 min at room temperature, and then, the material was filtered and added to tubes and placed in a water bath at 100 °C for 60 min. The tubes were cooled on ice, and the reading was performed using a spectrophotometer at 520 nm. Quantification was performed using a standard proline curve.

### 4.7. Statistical Analysis

The data were submitted to Shapiro–Wilk normality tests and a Barlett’s homogeneity of variance test, and when the assumptions were met, they were submitted to a two-way ANOVA with a post hoc Tukey test. Data were presented in bar graphs. When the data did not show normality or variance homogeneity, a rank transformation of the data was performed [67,68], and the data were represented in boxplots so that it was possible to better observe the data dispersion. A PCA was used to observe the multivariate correlation between all morphophysiological and biochemical variables and the treatment conditions. All statistical analyses and graphs were made with the R software environment using the tidyverse [69], multcomp [70], and rstatix [71] packages.

## 5. Conclusions

Exposure to I (as KI) increased tolerance to water deficit in soybean plants through modulation of the antioxidant enzymatic system and increased photosynthetic efficiency, which consequently provided a greater accumulation of biomass. In plants without water stress, KI concentrations greater than 10 μM induced toxic effects of this element, thus altering and increasing the root:shoot ratio as well as reducing the photosynthetic rate of soybean plants, whereas at higher concentrations (40 μM), it reduced the growth of soybean plants. Thus, based on the insights obtained for all the variables in conjoint using the PCA, the treatment with 10 μM I provided the best performance since it did not change biomass accumulation under adequate irrigation conditions, in addition to promoting tolerance of plants subjected to water deficit. However, further studies must be carried out to better elucidate the molecular, biochemical, and physiological processes induced by I in other plants grown under water deficit. In addition, studies must be carried out with the application of different sources and forms of I application to identify the best management for mitigating water deficiency in commercial production conditions, as well as carrying out cultivations with I application in soils to evaluate its possible interactions with other soil elements/components.

## Figures and Tables

**Figure 1 plants-12-02555-f001:**
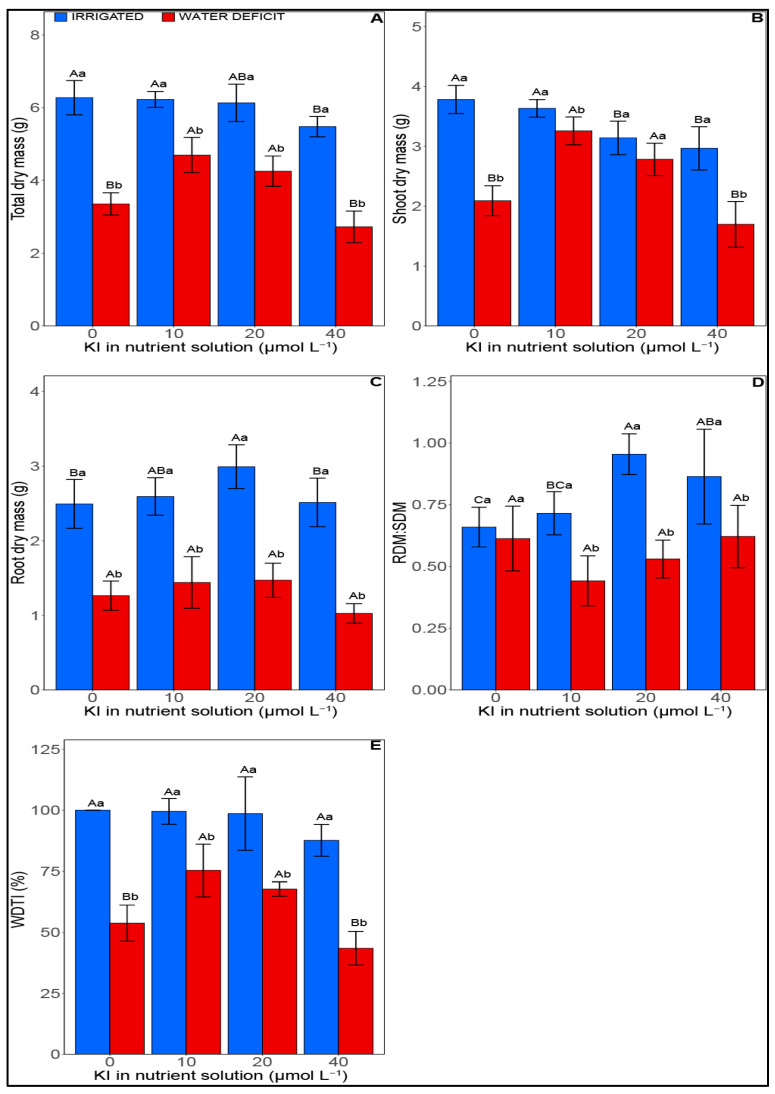
Effect of KI on growth attributes of soybean plants under water deficit or not. (**A**) total dry mass, (**B**) Shoot dry mass, (**C**) root dry mass, (**D**) RDM:SDM—shoot root relationship, and (**E**) WDTI—water deficit tolerance index. Irrigated samples are represented with blue bars, on the left, while samples with water deficit with red bars, on the right. Values are presented as average  ±  SD (*n* = 5). Equal letters indicate no significant differences (*p* > 0.05) calculated using the Tukey test. Equal uppercase letters indicate no distinction between KI concentrations, while equal lowercase letters indicate no distinction between the irrigation conditions (*p* > 0.05).

**Figure 2 plants-12-02555-f002:**
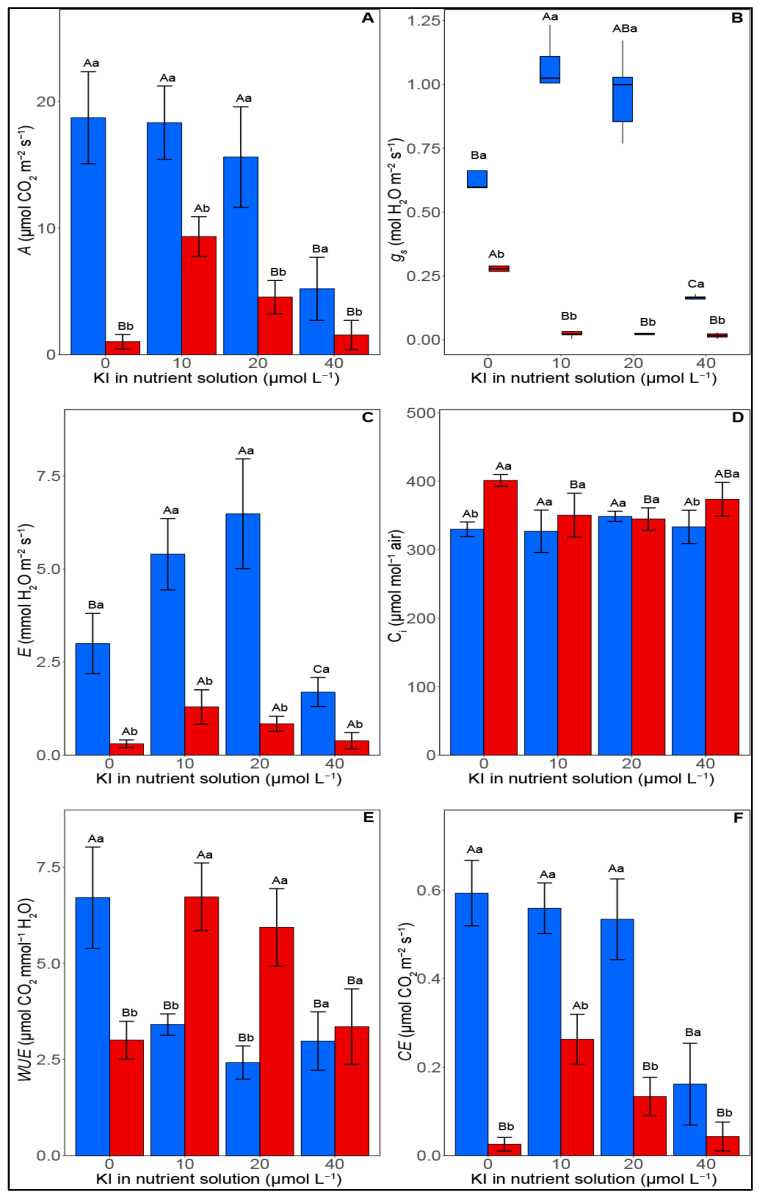
Effect of KI on gas exchange attributes of soybean plants under well-watered conditions (blue columns, on the left) and water deficit (red columns, on the right) one day after rehydration. (**A**) *A*—CO_2_ assimilation rate, (**B**) *g_s_*—stomatal conductance, (**C**) *E*—transpiration, (**D**) *C_i_*—internal CO_2_ concentration, (**E**) *WUE*—water use efficiency, and (**F**) *CE*—carboxylation efficiency. Values are presented as average  ±  SD (*n*  =  5). Equal letters indicate no significant differences calculated using the Tukey test (*p*  >  0.05). Different uppercase letters indicate differences between KI concentrations, while distinct lowercase letters indicate differences between irrigation conditions. Graphs represented in the boxplot indicate that the data did not meet the assumptions of normality and homogeneity of variance, requiring a transformation by rank.

**Figure 3 plants-12-02555-f003:**
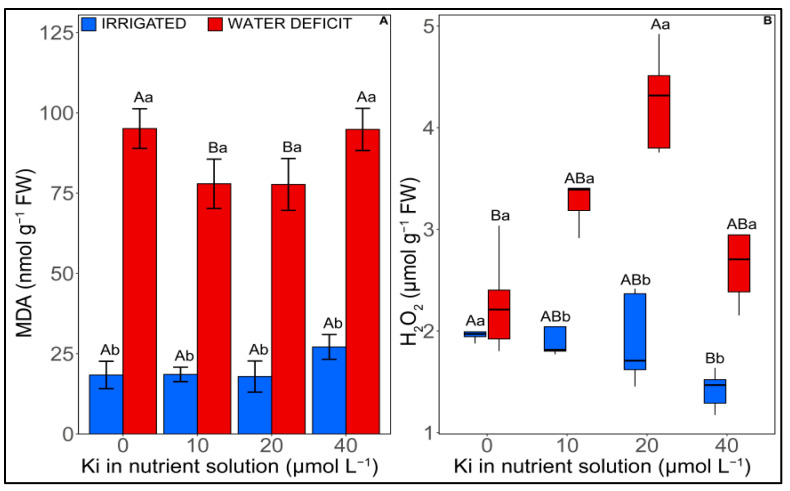
Effect of KI on oxidative damage attributes of soybean plants under well-watered conditions (blue columns, on the left) and water deficit (red columns, on the right) one day before rehydration. (**A**) MDA—malondialdehyde and (**B**) H_2_O_2_ —hydrogen peroxide. Values are presented as average  ±  SD (*n*  =  5). Equal letters indicate no significant differences calculated using the Tukey (*p*  >  0.05). Different uppercase letters represent statistical differences (*p* < 0.05) between KI concentrations, while lowercase letters indicate differences between irrigation conditions. Graphs represented in the boxplot indicate that the data did not meet the assumptions of normality and homogeneity of variance, requiring a transformation by rank.

**Figure 4 plants-12-02555-f004:**
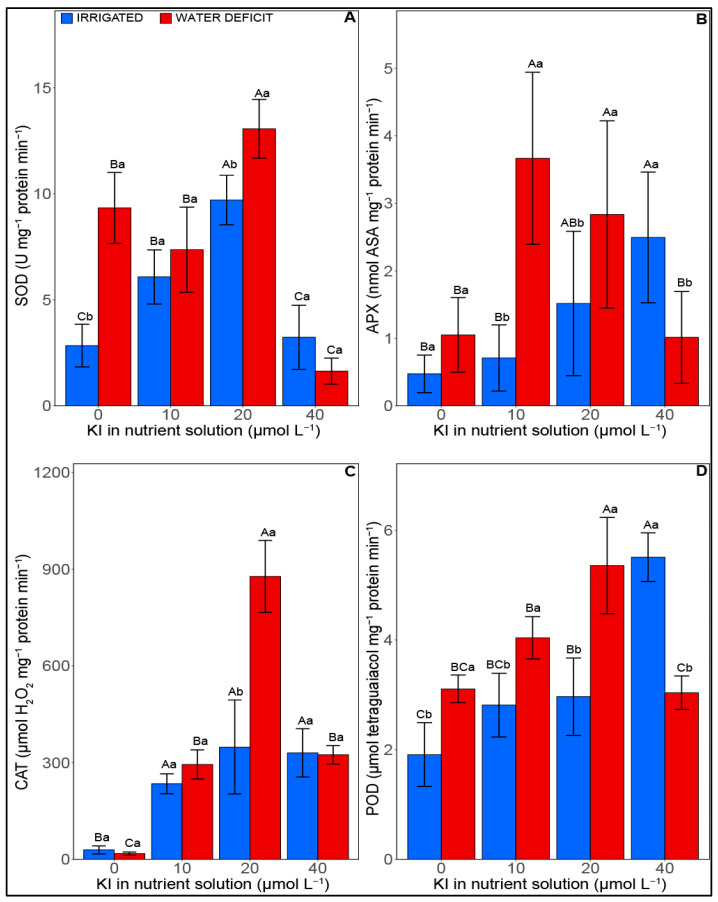
Effect of KI on antioxidant enzymatic activity attributes of soybean plants under well-watered conditions (blue columns, on the left) and water deficit (red columns, on the right) one day before rehydration. (**A**) (SOD—superoxide dismutase), (**B**) APX—ascorbate peroxidase, (**C**) CAT—catalase, and (**D**) POD—guaiacol peroxidase. Blue bars are irrigated samples and red bars are water deficit samples. Values are presented as average ± SD (*n*  =  5). Equal letters indicate no significant differences calculated using the Tukey test (*p*  >  0.05). Different uppercase letters represent statistical differences (*p* < 0.05) between KI concentrations, while lowercase letters indicate differences between irrigation conditions.

**Figure 5 plants-12-02555-f005:**
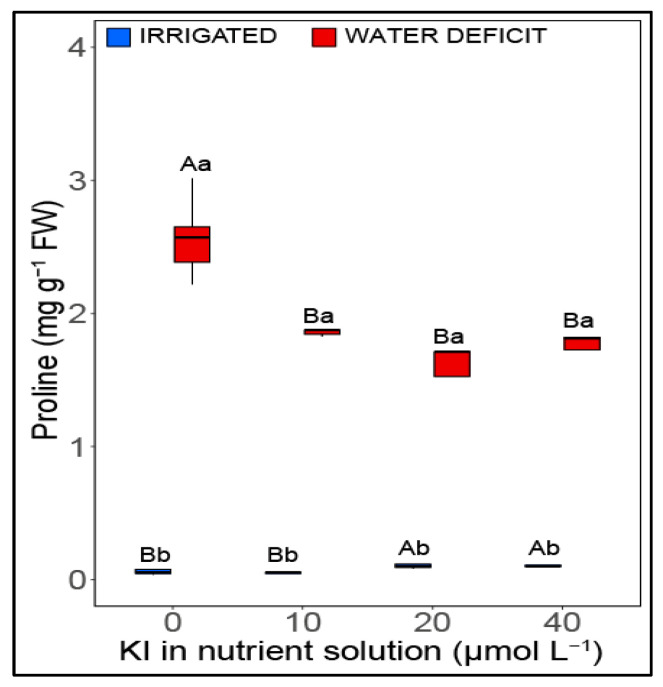
Effect of KI on content proline of soybean plants under well-watered conditions (blue columns, on the left) and water deficit (red columns, on the right) one day before rehydration. Values are presented as average ± SD (*n* = 5). Equal letters indicate no significant differences calculated using the Tukey test (*p*  >  0.05). Different uppercase letters represent statistical differences (*p* < 0.05) between KI concentrations, while lowercase letters indicate differences between irrigation conditions. Graphs represented in the boxplot indicate that the data did not meet the assumptions of normality and homogeneity of variance, requiring a transformation by rank.

**Figure 6 plants-12-02555-f006:**
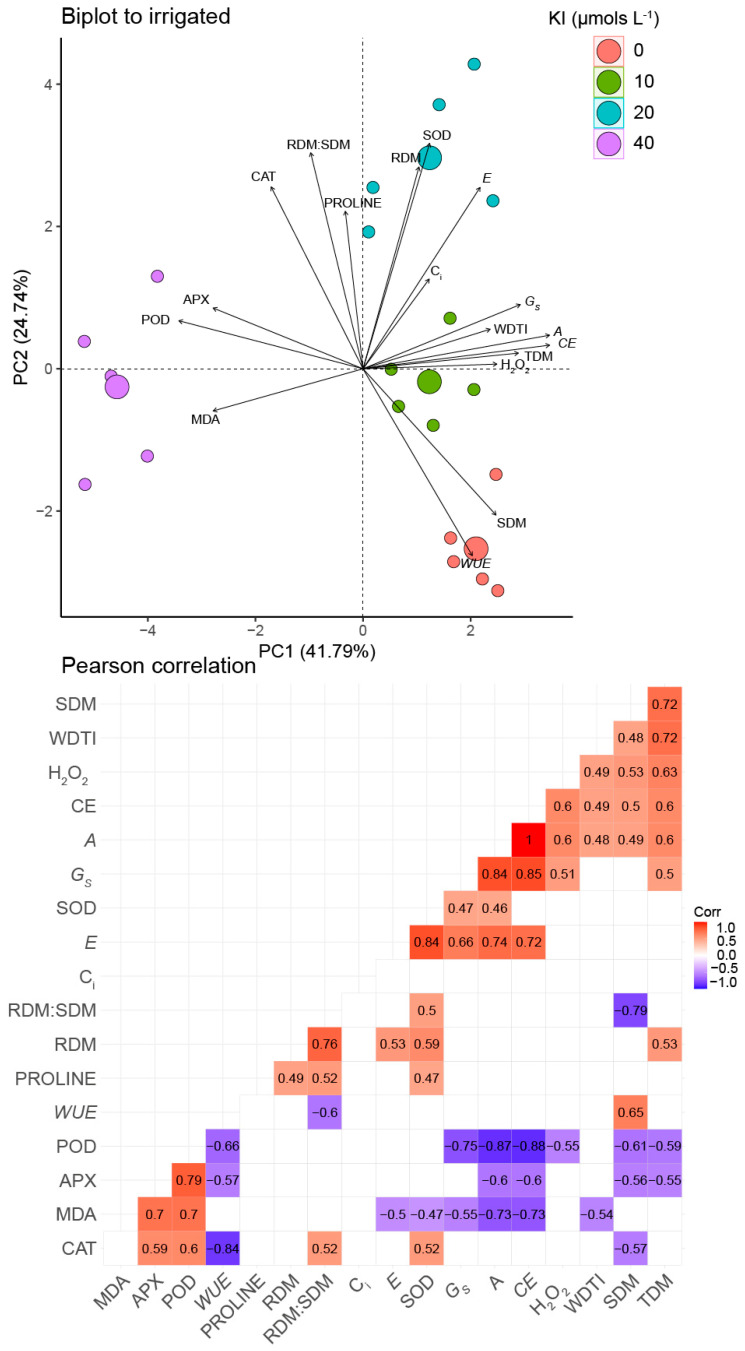
Principal component analysis and Pearson correlations using morphophysiological and biochemical data on soybean plants in response to well-watered conditions and KI application. Significant correlation coefficients (q < 0.01) are indicated with bold numbers, where positive and negative correlations are distinguished with red and blue, respectively. Non-significance is indicated with white boxes without numbers. TDM—total dry mass; SDM—shoot dry mass; RDM—root dry mass; WDTI—water index tolerance; H_2_O_2_—hydrogen peroxide; MDA—malondialdehyde; *A*—CO_2_ assimilation rate; *g_s_*—stomatal conductance; *C_i_*—internal concentration of CO_2_; *E*—transpiration; *WUE*—water use efficiency; *CE*—carboxylation efficiency; POD—guaiacol peroxidase; APX—ascorbate peroxidase; CAT—catalase.

**Figure 7 plants-12-02555-f007:**
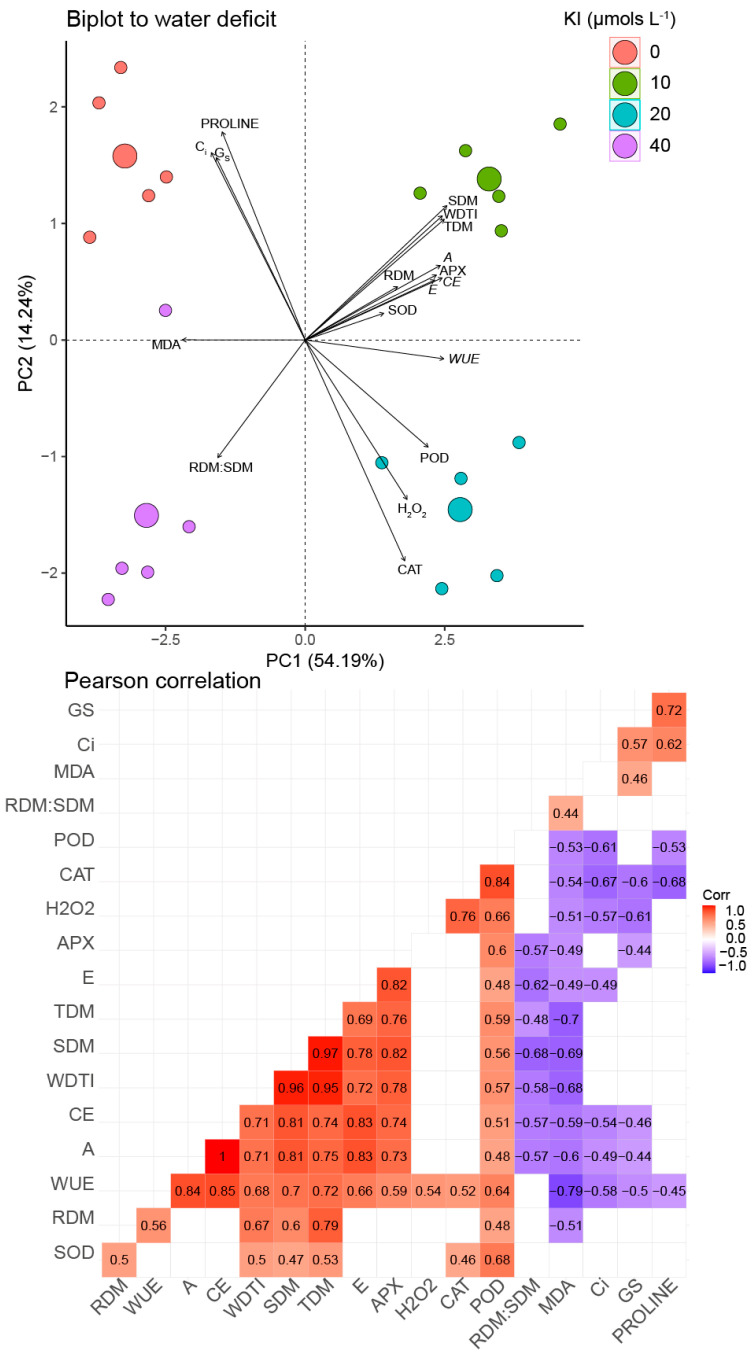
Principal component analysis and Pearson correlations using morphophysiological and biochemical data on soybean plants in response to water deficit and KI application. Significant correlation coefficients (q < 0.01) are indicated with bold numbers, where positive and negative correlations are distinguished with red and blue, respectively. Non-significance is indicated with white boxes without numbers. TDM—total dry mass; SDM—shoot dry mass; RDM—root dry mass; WDTI—water index tolerance; H_2_O_2_—hydrogen peroxide; MDA—malondialdehyde; *A*—CO_2_ assimilation rate; *g_s_*—stomatal conductance; *C_i_*—internal concentration of CO_2_; *E*—transpiration; *WUE*—water use efficiency; *CE*—carboxylation efficiency; POD—guaiacol peroxidase; APX—ascorbate peroxidase; CAT—catalase.

**Figure 8 plants-12-02555-f008:**
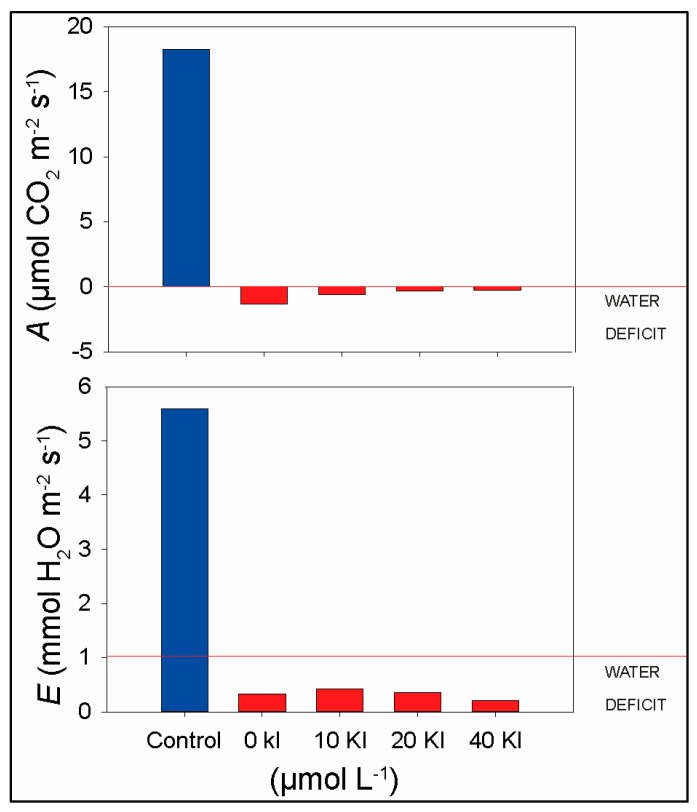
Establishment of water deficit by monitoring gas exchange in soybean plants subjected to different concentrations of KI. The graph shows the mean values of *A* and *E* on the fourth day after the suspension of irrigation.

**Table 1 plants-12-02555-t001:** Mean values of A and E for monitoring the water deficit of soybean plants exposed to different concentrations of KI.

TREATMENTS	*A* (μmol CO_2_ m^−2^ s^−1^)	*E* (mmol H_2_O m^−2^ s^−1^)
CONTROL	18.23	5.58
WATER DEFICIT + 0 μM KI	−1.26	0.32
WATER DEFICIT + 10 μM KI	−0.58	0.41
WATER DEFICIT + 20 μM KI	−0.28	0.35
WATER DEFICIT + 40 μM KI	−0.26	0.20

## Data Availability

Not applicable.

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
