# Peer review of "Soybean Plants Exposed to Low Concentrations of Potassium Iodide Have Better Tolerance to Water Deficit through the Antioxidant Enzymatic System and Photosynthesis Modulation"

_plants, 2023, doi:10.3390/plants12132555_

Round 1

Reviewer 1 Report

The present manuscripts describes the impact of applying Iodine on the growth of soybean and the benefits to resist short hydric stresses in this plants.

The introduction is a complete review of the topic in the litterature, and presents the objectif of the study clearly.

The results are clearly presented and well statistically analysed. It provides a good dicussion of the results and the conclusion is pertinent.

The material and methods is well presented and clearly described.

Only minor form corrections should be done, please find the localisation in the attached file.

Author Response

Dear reviewer, thank you for your attention and contribution to our work. All suggested changes have been made.

Reviewer 2 Report

In this study, authors studied the effects of different concentrations of potassium iodide (KI) on the antioxidant defense system of plants. The results showed exposure to lower I concentrations improved gas exchange and reduced lipid peroxidation; also it was observed that greater concentrations of I promoted negative changes in photosynthetic efficiency, as well as in biomass accumulation and partition. However, the hypothesis and innovativeness showed be strengthened as there are so much similar studies on other plants. Besides, I was confused about the results in some of the figures. For example:

1.Fig.4 it is strange that at 20 umol L-1, why both POD and CAT (also H2O2?) increased when SOD increased?

2. Fig.5 why pro was not increased at 20 umol L-1?

3. 4.1 section: Is there any size of seedlings before treatment? Are there any photos about it?

Author Response

Dear reviewer, thank you for your attention and contribution to our work. Below are the detailed points of the suggested revisions:

  1. It is strange that at 20 μmol L-1, why both POD and CAT (also H2O2?) increased when SOD increased?

Answer: Dear reviewer, your question is very relevant. One of the main processes proposed for iodine actions in plants reported by some published papers is the induction of an increase in ROS, including H2O2, to stimulate an increase in the activity of antioxidant enzymes (Kuepper et al. 2008; Luther et al. 1995; Medrano-Macías et al. 2018). This seems to be a signaling effect, which prepares the plants for later stresses. So, the high enzymatic antioxidant activity at the concentration of 20 μM KI seems to be related to the maintenance of cellular homeostasis with normal H2O2 levels, as mentioned in the Discussion section. Although H2O2 levels are high at a concentration of 20 μM KI, they are within those proposed by Poller (2001) and Mittler (2002) for basal levels of H2O2, in some cell compartments, such as chloroplasts. Thus, we believe that this level of H2O2 presented is related to cell signaling and not to oxidative stress. Through the uni- and multivariate analyses carried out in our work, it is possible to notice that the concentration of 20 μM KI does not have a positive relationship with the MDA content, unlike the plants subjected to water deficit without treatment with KI. Thus, we believe that this high enzymatic activity at 20 μM KI led to the lower conversion of H2O2 into -OH, and consequently lower lipid peroxidation, represented by the MDA content. On the other hand, the lower concentration of H2O2 in plants 0 μm KI and subjected to water deficit was due to the conversion of H2O2 into -OH, consequently resulting in greater MDA, since in the uni- and multivariate analyzes this treatment favored the accumulation of MDA. This explanation is in the discussion of our paper (see p. 15-16, lines 345-363).

Kuepper, F.C.; Carpenter, L.J.; McFiggans, G.B.; Palmer, C.J.; Waite, T.J.; Boneberg, E.-M.; Woitsch, S.; Weiller, M.; Abela, R.; Grolimund, D.; et al. Iodide Accumulation Provides Kelp with an Inorganic Antioxidant Impacting Atmospheric Chemistry. Proceedings of the National Academy of Sciences of the United States of America 2008, 105, 6954–6958, doi:10.1073/pnas.0709959105.

Luther, G.W.I.; Wu, J.; Cullen, J.B. Redox Chemistry of Iodine in Seawater. In Aquatic Chemistry; Advances in Chemistry; American Chemical Society, 1995; Vol. 244, pp. 135–155 ISBN 978-0-8412-2921-1.

Medrano-Macías, J.; Mendoza-Villarreal, R.; Robledo-Torres, V.; Fuentes-Lara, L.O.; Ramírez-Godina, F.; Pérez-Rodríguez, M.Á.; Benavides- Mendoza, A.; Medrano-Macías, J.; Mendoza-Villarreal, R.; Robledo-Torres, V.; et al. The Use of Iodine, Selenium, and Silicon in Plant Nutrition for the Increase of Antioxidants in Fruits and Vegetables. In Antioxidants in Foods and Its Applications; IntechOpen, 2018 ISBN 978-1-78923-379-7.

Polle, A. Dissecting the Superoxide Dismutase-Ascorbate-Glutathione-Pathway in Chloroplasts by Metabolic Modeling. Computer Simulations as a Step towards Flux Analysis. Plant Physiol 2001, 126, 445–462, doi:10.1104/pp.126.1.445.

Mittler, R. Oxidative Stress, Antioxidants and Stress Tolerance. Trends in Plant Science 2002, 7, 405–410, doi:10.1016/S1360-1385(02)02312-9.

  1. Why pro was not increased at 20 μmol L-1?

Answer: We believe that proline is an important compatible solute for mitigating water deficit in plants. In figure 5 it is possible to observe that all plants subjected to water deficit had the proline content increased when compared with plants receiving adequate irrigation. However, what did not alter the proline content was the application of iodine. We, therefore, understand that this element may be more related to antioxidant responses in plants, as also suggested in previous work.

  1. 1 section: Is there any size of seedlings before treatment? Are there any photos about it?

Answer: We don’t have photos of these plants: however, we understand that this information does not compromise the dataset of our manuscript. The size of the seedlings used in the experiment was standardized to the maximum and the treatments were randomly distributed, as well as there were enough repetitions to prove our results. In addition, we added information about the vegetative stage of the seedlings in the methodology (see line 411).

Finally, we would like to answer to the comment: "However, the hypothesis and innovativeness showed to be strengthened as there are so much similar studies on other plants."

Answer: As mentioned in the cover letter, this is one of the first manuscripts that show the efficiency of iodine (with potassium iodide as a source) in mitigating water deficit. Other existing manuscripts in the literature include other types of stress, such as salinity and heavy metals. We found only one (very recent) paper that deals directly with this topic (Jafarian et al. 2020). However, it’s important to mention that this aforementioned paper involves the iodine effect only on germination and the initial growth of plants. On the other hand, our manuscript focused on young plants (vegetative phase) and includes several physiological and metabolic responses induced by iodine in plants submitted to water-deficiency conditions. Moreover, we add this justification in the Introduction section of our work (see lines 90-93).

Reviewer 3 Report

The manuscript deals with the soybean response to water deficit in terms of exposure to potassium iodide. The paper is of scientific and practical importance due to the global problem of water deficit and the development of new strategies for plant adaptation to unfavorable climatic conditions. However, I have some comments listed below:

L25-27: indicate consequently whether KI or I was applied in this study. Unify throughout the paper

L26-31: add some % changes of examined parameters between control and treatments

L71-74: firstly indicate the ubiquitous role of antioxidant enzymes in mitigating different abiotic stresses (climatic conditions, pesticides, heavy metals). For this purpose the Authors may refer to the following reference: https://doi.org/10.1016/j.scienta.2022.110988

L106: the highest root:shoot ratio was noticed with 20 µM in Fig. 1

L154: in Fig. 2 D, E, F max KI concentration is 50 µM, while in others 40 µM

L157: unify the meaning of the bars. Red column indicate water deficit, while blue one irrigated condition. This mistake is also shown in Fig. 3, 4, 5. Indicate the meaning of blue and red bar directly on the Figure as it is in Fig. 1

L257 and L276: add a scale with colors and numbers referring to the Pearson correlation. Why are some correlation coefficients not included and shown as empty?

L259: ‘response to water deficit’ – rather, it refers to irrigated treatments

L390: indicate pot dimensions. How many plants were cultivated in one pot?

L405: describe A and E value

Author Response

Dear reviewer, thank you for your attention and contribution to our work. Below are the detailed points of the suggested revisions:

  1. L25-27: indicate consequently whether KI or I was applied in this study. Unify throughout the paper.

Answer: The issue has been clarified.

  1. L71-74: firstly indicate the ubiquitous role of antioxidant enzymes in mitigating different abiotic stresses (climatic conditions, pesticides, heavy metals). For this purpose the Authors may refer to the following reference: https://doi.org/10.1016/j.scienta.2022.110988.

Answer: The issue has been clarified.

  1. L106: the highest root:shoot ratio was noticed with 20 µM in Fig. 1

Answer: The issue has been clarified.

  1. L154: in Fig. 2 D, E, F max KI concentration is 50 µM, while in others 40 µM.

Answer: The issue has been clarified.

  1. L157: unify the meaning of the bars. Red column indicate water deficit, while blue one irrigated condition. This mistake is also shown in Fig. 3, 4, 5. Indicate the meaning of blue and red bar directly on the Figure as it is in Fig. 1.

Answer: The issue has been clarified.

  1. L257 and L276: add a scale with colors and numbers referring to the Pearson correlation. Why are some correlation coefficients not included and shown as empty?

Answer: The issue has been clarified. Spaces represent non-significant correlation between variables, this information was added to the captioned figure.

  1. L259: ‘response to water deficit’ – rather, it refers to irrigated treatments.

Answer: The issue has been clarified.

  1. L390: indicate pot dimensions. How many plants were cultivated in one pot?

Answer: The issue has been clarified. The capacity of the pot was incorrectly informed, i.e., the pot had a capacity of 1000g and the information was corrected in the text.

  1. L390: indicate pot dimensions. How many plants were cultivated in one pot?

Answer: The issue has been clarified.

Round 2

Reviewer 2 Report

Modifications  have been made.

Author Response

Thanks

Reviewer 3 Report

The Authors have corrected the manuscript according to the comments. However, in L71-76, the background of antioxidant system description is insufficient to highlight the importance of this mechanism related to the content of this study. Please describe the ubiquitous role of antioxidant enzymes in mitigating different abiotic stresses (climatic conditions, pesticides, heavy metals). For this purpose the Authors may refer to the following reference: https://doi.org/10.1016/j.scienta.2022.110988

Author Response

Dear reviewer, thank you for your contribution. This is a well-described mechanism that has been confirmed in several studies. Furthermore, we have well described how these enzymes play a role in the cellular detoxification process by ROS (lines 72-78). We understand that the main focus of our manuscript is whether iodine can potentiate these responses in plants under water deficit, as it occurs in other types of situations, which is described in our text (Lines 78-82) and confirmed by our results. However, we understand that the suggestions made go towards adding references that emphasize this role of antioxidant enzymes in stress. Thus, we have added quotes from works that support our descriptions.

Round 3

Reviewer 3 Report

It can be stated that the Authors have corrected the manuscript. I have no more comments.